# Emulsified Homo (Ciprofloxacin) Polymer Nanoparticles for Antibacterial Applications

**DOI:** 10.3390/ijms262010034

**Published:** 2025-10-15

**Authors:** Faeez Mahzamani, Edward Turos

**Affiliations:** Department of Chemistry, University of South Florida, 4202 E. Fowler Avenue, Tampa, FL 33620, USA; fmahzama@usf.edu

**Keywords:** nanoparticles, ciprofloxacin, antibacterial, emulsion polymerization, *Staphylococcus aureus*

## Abstract

We report for the first time a method for forming polyacrylate nanoparticles using N-acryloylciprofloxacin as a sole monomer for emulsion polymerization. The procedure involves a free radical-induced emulsion polymerization of N-acryloylciprofloxacin monomer to produce a stable aqueous emulsion comprising uniformly sized polyacrylate nanoparticles. Dynamic light scattering analysis of the emulsions showed a single population of nanoparticles having an average diameter of 970 nm and average surface charge of −63 mV, indicative of the high stability of the emulsion and significantly enhance lipophilicity of the polymeric matrix of the nanoparticle. Antibacterial testing of the emulsions against the Gram-positive microbe *Staphylococcus aureus* and the Gram-negative *Escherichia coli* found in vitro activities identical to those of the reference clinical agent, ciprofloxacin. Assays against human colorectal carcinoma cells and human embryonic kidney cells showed essentially no cytotoxicity. This is the first study on the synthesis of aqueous nanoparticle emulsions assembled solely from a single monomer derived from the antibiotic agent.

## 1. Introduction

Previous experiments in our laboratory have demonstrated the ability to form aqueous polyacrylate nanoparticle emulsions for the purpose of water-solubilizing and encasing certain antibacterial compounds, as a means to improve their stability and antibiotic activity especially towards multi-drug-resistant strains of bacteria [1,2]. These nanoparticle emulsions were prepared by radical-induced emulsion polymerization of butyl acrylate/styrene mixtures (7:3 *w*/*w*) in water at 60 °C using sodium dodecyl sulfate (SDS) as an emulsifying agent and potassium persulfate as a radical initiator (Figure 1).

The reactions led to the formation of homogeneous, stable aqueous emulsions containing uniformly sized nanoparticles of 45–50 nm in diameter. The method was successfully applied to penicillins and N-thiolated β-lactams, such that the antibacterial agents could be introduced into the nanoparticle either by non-covalent entrapment as a free drug, or covalently via an N-acryloyl derivative. The antibiotic-containing nanoparticles showed promising in vitro activity against pathogenic bacteria such as methicillin-resistant *Staphylococcus aureus* (MRSA).

While these earlier nanoparticle emulsions provided increased water solubility and, in some cases, improved bioactivity of the β-lactam antibacterial agent, the polyacrylate backbone largely comprised non-bioactive monomers (butyl acrylate–styrene or methyl methacrylate–styrene), and only 1–3% (by weight) of the antibacterial agent in the nanoparticle. The amount of drug loading into the nanoparticle during the assembly process was limited by how much surfactant could be used, given that amounts exceeding 3% (by weight) of SDS caused discernable cytotoxicity. The emulsions containing up to 20% of solid content (as a mixture of nanoparticles and a small amount of non-emulsified polymers) in fact carried 0.2–1% of the antibacterial agent inside of the nanoparticles. The emulsions were typically milky in consistency and somewhat sticky when exposed to air, causing films to rapidly form when dried, unwanted coagulation within syringes, micro-porous filters, and gel columns that made it very difficult to purify and use for in vivo testing. We were able to overcome some of these issues with purification techniques to remove residual unreacted monomers and non-surfactant-stabilized oligomers within the emulsion and the use of other acrylate monomers [3,4] and surfactant combinations to try to enhance the amount of antibiotic that could be entrapped, or to alter nanoparticle sizes, without increasing overall cytotoxicity or instability of the emulsion.

In this report, we describe an altogether new approach to preparing antibiotic-bound polyacrylate emulsions that completely obviates the restriction of using butyl acrylate and styrene (or other monomers) to form the polymeric particle framework, and the resulting limitations those extraneous monomers place on the properties of the nanoparticles. The new procedure uses the antibiotic compound itself as the sole acrylate monomer for the polymerization, a technique that to our knowledge has never been previously reported and may thus be an important advance in the antibacterial organopolymer field. Moreover, we confirm that the antibacterial agent, ciprofloxacin, retains its antibacterial capabilities despite being incorporated into the polymer matrix.

## 2. Results

### 2.1. Synthesis of Nanoparticle Emulsions

For our studies, we chose ciprofloxacin as the platform antibiotic for the formation of the polyacrylate nanoparticles. The N-acryloyl derivative of commercial ciprofloxacin hydrochloride was prepared for this purpose according to our previously reported acylation procedure (Figure 2) [5].

The polyacrylate emulsions were prepared using a modified protocol that we previously reported (Figure 3) [1,2]. The full experimental details can be found in the Materials and Methods section.

### 2.2. Characterization of the Physical Properties of the Nanoparticle Emulsions

In analyzing the resulting emulsions, the first question we hoped to address was whether any nanoparticles were being formed in the emulsion polymerization process. For this, we used dynamic light scattering analysis to evaluate the average size and surface charge of the emulsion using a Malvern Zetasizer nano-ZS instrument (Malvern Panalytical, Westborough, MA, USA). Each sample was analyzed in triplicate as described in the Materials and Methods section. The size distribution shows a single narrow peak indicating the uniformity of the emulsion with a single population centered on average at approximately 970 nm. Similarly, surface charge measurements indicated a highly stable emulsion, with an average reading of −63 (±5.6) mV.

### 2.3. Microbiological Testing of the Nanoparticle Emulsions

To investigate whether the nanoparticles possess antibiotic capabilities, each crude emulsion was tested against *Staphylococcus aureus* (ATCC 25923) and *Escherichia coli* (K12) using a 96-well plate broth assay to determine the minimum inhibitory concentration (MIC). The procedure is described in the Materials and Methods section. The MIC determination was conducted in triplicates for each bacterium, with ciprofloxacin hydrochloride being used as a positive control and a blank broth used as a negative control. The MIC was the lowest concentration of the antibiotic that completely inhibited bacterial growth (visually) within that series of dilutions.

### 2.4. Cell Cytotoxicity Assays of the Nanoparticle Emulsions

In vitro cell cytotoxicity was tested on two human cell lines, human colorectal tumor cells HCT-116, and human embryonic kidney cells HEK 293. The test emulsion was diluted using the complete growth medium for each cell type and added into the wells of each test plate to give a final concentration of N-acryloylciprofloxacin of 2 mg/mL, 1 mg/mL, 0.5 mg/mL, 0.25 mg/mL, 0.125 mg/mL, and 0.0625 mg/mL within a series. The testing was performed in triplicate, and one well in each triplicate was left untreated as the negative control for 100% growth. The plates were incubated for 48 hours at 37 °C under an atmosphere of 5% CO_2_, then a 5 mg/mL solution of 3-(4,5-dimethyl-2-thiazolyl)-2,5-diphenyltetrazolium bromide (MTT) in sterile phosphate-buffered saline was added to give a 10% final concentration in each well. The plates were then further incubated for 4 hours at 37 °C under an atmosphere of 5% CO_2_ and the IC_50_ was determined as the well with at least 50% cell viability compared to the untreated control cell with 100% cell growth. 

## 3. Discussion

Poly(N-acryloylciprofloxacin) nanoparticle emulsions were successfully prepared by modification of the previously reported emulsion polymerization methodology. The main difference with this new procedure was the pre-solubilization of the water-insoluble antibacterial agent in an organic solvent to permit more uniform addition into the aqueous solution, as a means to form homogeneous emulsions. We found that dichloromethane provided the best combination of solubilizing the ciprofloxacin monomer and being volatile enough to evaporate from the media during emulsion polymerization. We also found that the increased temperature (90 °C rather than 60 °C), stir speed (1100 rpm rather than the usual 750 rpm), and the addition of sodium dodecyl sulfate before the monomers were added provided more optimal results. Additionally, it was advantageous to allow the reactions to run for 48 hours rather than the usual 6 hours required for the butyl acrylate–styrene co-monomer systems.

Dynamic light scattering (DLS) analysis of the crude emulsions confirmed a major population of nanoparticles measuring approximately 970 nm in diameter (Figure 4). In addition, the particles possessed an average zeta potential of −63 (± 5.6) mV, indicative of high stability as an aqueous emulsion. Indeed, emulsified samples stored at room temperature have remained completely viable and without significant changes in the DLS data after many months in glass vials. It is notable that these homo(N-acryloylciprofloxacin) nanoparticles are much larger than those previously constructed with butyl acrylate–styrene co-monomers, which routinely measured 45–50 nm in diameter. The basis for this 20-fold increase in size is hypothesized to reflect the increasing hydrophilicity of the poly(N-acryloylciprofloxacin) nanoparticle matrix, compared to the highly lipophilic butyl acrylate/styrene polymer chain of the previous emulsified nanoparticles that disfavors the influx of water from the surrounding aqueous environment. Thus, increasing the concentration of the monomer into the polymerization protocol increases the degree in which the resulting nanoparticle matrix draws in water. We saw a general trend of increasing nanoparticle size as the amount of N-acryloylciprofloxacin was increased from 1% to 10 weight % in forming the polymer emulsions. This is an identical phenomenon with what we previously observed for polyacrylate nanoparticles constructed of hydrophilic glycosylated acrylate monomers, versus those having highly lipophilic hydroxyl-protected glycosylated acrylate monomers [3,4]. These DLS data afford the average sizes and size distributions of the emulsified particles in aqueous media. We also attempted to evaluate particle sizes by tunneling electron microscopy and scanning electron microscopy; however, these efforts were thwarted by the deposition of undefined masses rather than individual particles on the wafer surface, even under very high dilution of the emulsion.

Our laboratory has reported on the use of electron microscopies, namely SEM and TEM, to examine morphology and dimensions of emulsified polyacrylate nanoparticles [2]. The use of electron microscopies requires the preparation of completely desiccated emulsions applied directly onto inorganic support wafers. While we were successful at preparing samples of polyacrylate nanoparticles derived from butyl acrylate and styrene, as previously reported, to confirm the individual particle sizes closely matched those observed by DLS for the emulsified solutions, these, and high dilution of the emulsions followed by vacuum desiccation during sample prep gave no intact particles by TEM. We believe this data indicates that the new emulsified poly(N-acryloylciprofloxacin) particles appear to be highly gelatinous and whose overall structures are more dependent on the stabilizing effects of the surfactant and aqueous media. These differences, and their larger sizes, may inhibit the ability to penetrate into human cells and tissues and could make the particles susceptible to removal by the lymphatic system before they can effectively control infection. Thus, at the present, we can only conclude that these emulsified poly(N-acryloylciprofloxacins have potent antibacterial activity equal to that of ciprofloxacin towards *S. aureus* and *E. coli*, but we cannot yet say if these are to be useful for drug delivery applications.

In vitro antibacterial studies confirmed that the nanoparticle emulsions are strongly bioactive, with an MIC of 0.5 µg/mL for *S. aureus* and 0.012 µg/mL against *E. coli* (Figure 5).

These values are identical to those of ciprofloxacin, the control antibiotic. The finding that these nanoparticle polymers show any antibacterial capabilities at all was not totally expected, given the large dimensions of the particles, and the fact that the active antibacterial agent is most likely to be chemically attached to the interior of the nanoparticle matrix. As an antibiotic, ciprofloxacin must enter the bacterial cell to arrive at its cellular target, bacterial DNA gyrase, and bind within the gyrase tertiary structure. Attachment of ciprofloxacin to the polymer backbone of the nanoparticle presumably requires hydrolysis of the amide linkage, prior to interaction of the released antibiotic with DNA gyrase. This may occur either outside of the cell, or within the bacterium itself, most likely through enzymatic involvement, as the amide functionality is a difficult one to hydrolyze otherwise. The in vitro cytotoxicity results for both human colorectal carcinoma cells HCT-116 and human embryonic kidney cells HEK-293 were also informative. The observed IC_50_ was 500 µg/mL for both cell lines, a 1000-fold difference over the bacterial MIC value for *S. aureus* and greater than 40,000-fold for *E. coli* (Figure 6).

We note that these data are preliminary, and more detailed analyses are required for evaluating potential susceptibilities and toxic effects, if any, in appropriate rodent models and additional human cell lines. That will need to be examined.

Lyophilization of the nanoparticle emulsion produced an amorphous powder that unfortunately could not be reformulated back to its original emulsified state through addition of water, even with vigorous agitation or ultrasonic trituration. Moreover, the lyophilized powder remained completely insoluble in common organic solvents including methanol, ethanol, dichloromethane, hexane, acetone, ethyl acetate, and dimethylformamide. We believe this suggests that once the structural integrity and morphology of the surfactant-stabilized nanoparticles is compromised, reconstitution of the desiccated particles to an emulsified state is irreversibly lost. We did note that extraction of the solid material with methanol, ethanol, dichloromethane, hexane, acetone, and ethyl acetate failed to show any trace of unreacted N-acryloylciprofloxacin monomer upon evaporation and analysis by proton NMR spectroscopy. This confirms that the emulsion polymerization is complete, and thus, all of the ciprofloxacin monomer is incorporated into the framework of the nanoparticle. Attempts to perform the emulsion polymerization procedure on the non-acryloylated ciprofloxacin or the N-acetylciprofloxacin analog led to a bilayer mixture, not an emulsion, with the layers separating within seconds after stirring was stopped. Therefore, the acryloyl group is a clear prerequisite for nanoparticle formation.

Though the field of antibacterial polymers is well-explored, typically, the preparation of these materials requires the use of co-monomers in organic media. Recent advances in the preparation of emulsified or otherwise suspended antibacterial agents for intracellular delivery have been reported [6,7,8,9,10,11,12,13,14,15,16,17,18,19,20,21]. As far as we are aware, this is the first example of an *aqueous* nanoparticle polymer emulsion being formed from a single monomer that carries the antibiotic agent itself. It is also a demonstration that the emulsification by radical polymerization in aqueous media can be executed efficiently with an antibacterial acrylamide. We hope to further investigate the properties of these N-acryloylciprofloxocin nanoparticle emulsions and to further expand the methodology to other antibiotic agents. We believe that the data presented here on the synthesis and characterization of these emulsified poly(*N*-acryloylciprofloxacin) particles suggest that while they may not be immediately useful for treating systemic infections in humans, or as drug delivery platforms, they are indeed most unique and may have utility in other applications in the antibacterial materials domain. The significance of this study, however, was to demonstrate for the first time that polymeric particles made of a single antibacterial monomer within their composition can be formed in aqueous media by emulsion polymerization and are stable emulsions with antibacterial capabilities.

## 4. Materials and Methods

Synthesis of N-acryloylciprofloxacin: To a round bottom flask, 120 mL of dichloromethane was added, then 3.0 g (9.0 mmol) of ciprofloxacin and 1.9 mL (13.5 mmol) of triethylamine. The mixture was left to stir at 0 °C for 1 h, then acryloyl chloride (1.1 mL, 13 mmol) was added dropwise. The ice bath was removed and the reaction was left to stir overnight. The solution was added dropwise to a flask containing hexane (60–80 mL) and the resulting solid was collected by filtration and allowed to air dry. The resulting yield was 2.90 g (84%) as a pale yellow solid. Melting point exceeded 250 °C. ^1^H NMR (400 MHz, CDCl_3_) δ 1.18 (br. s., 2 H), 1.38 (d, J = 6.6 Hz, 2 H), 3.33 (m, 4 H), 3.51 (br. s., 1 H), 3.47 (m, 1 H), 3.86 (m, 4 H), 5.76 (dd, J = 10.5, 1.7 Hz, 1 H), 6.35 (dd, J = 16.8, 1.7 Hz, 1 H), 6.59 (dd, J = 16.8, 10.5 Hz, 1 H), 7.35 (d, J = 7.1 Hz, 1 H), 8.03 (d, J = 12.8 Hz, 1 H), 8.75 (s, 1 H).

The method for forming the poly (N-acryloylciprofloxacin) emulsion requires the following procedure: to a round bottom flask, 4 mL of deionized water was added, which was then stirred using a 1.25 cm (300 mg) Teflon-coated magnetic stir bar at 1000 rpm on a Corning PC-420D magnetic stirrer at 30 °C using a self-regulated oil bath. To this, 30 mg of SDS was added. N-Acryloylciprofloxacin (500 mg) was dissolved in 1 mL of warm dichloromethane, and this solution was added dropwise to the deionized water–SDS mixture. A vent was placed on top of the flask by inserting a small stainless steel syringe needle through a rubber septum on the flask, under dry nitrogen, and the temperature of the mixture was increased at a rate of 5 °C per 30 min until reaching 90 °C. The mixture was left to stir overnight at this temperature, under an atmosphere of dry nitrogen. Potassium persulfate (10 mg) was added with an additional 0.5 mL of deionized water, and the mixture was stirred for 24 h, then removed from the oil bath. When cooled to room temperature, the resulting emulsion was decanted into a storage vial for dynamic light scattering analysis and biological testing.

Dynamic light scattering analysis was performed using a Malvern Zetasizer nano-ZS instrument (Malvern Panalytical, Westborough, MA, USA). To prepare the samples for the analyses, the freshly prepared emulsion was subjected to centrifugation at 10,000 rpm for 5 min using an Eppendorf Centrifuge 5424 (Eppendorf SE, Hamburg, Germany). An aliquot of the liquid emulsion was then drawn and deposited into a Malvern disposable folded capillary cell DTS-1070. Each sample was analyzed in triplicate, and each data collection consisted of 1 run of 100 scans (for size analysis) and three runs of 100 scans (for zeta potential determination). The size distribution shows a single narrow peak indicating the uniformity of the emulsion with a single population centered on average at approximately 970 nm. Similarly, surface charge measurements indicated a highly stable emulsion, with an average reading of −63 (±5.6) mV.

For microbiological testing, the original stock emulsion was diluted using the above Trypticase Soy Broth (TSB) solution to an initial concentration of 1.28 mg/mL of the N-acryloylciprofloxacin, then serial diluted with TSB to half the concentration each time. A volume of 10 µL of each emulsion dilution was added to a well in series, resulting in a final concentration run of 64 µg/mL to 0.012 µg/mL. The MIC determination was conducted in triplicates for each bacterium, with ciprofloxacin hydrochloride being used as a positive control and a blank broth used as a negative control.To prepare the bacteria for culture, all solutions were autoclaved prior to use. The bacteria were grown overnight at 37 °C on an agar plate composed of BBL TSA II Trypticase Soy Agar (TSA) and BBL Trypticase Soy Broth (TSB) in a 1:2 ratio at 4.4% concentration. A broth solution of 2.4% TSB was inoculated using the bacteria from the agar plates and was incubated at 37 °C to reach a 0.5 McFarland standard. The bacteria were then further diluted by a factor of 1000 using a broth solution of 2.4% TSB, and 190 µL of the diluted bacterial solution was transferred by micropipette into each well. The inoculated plates were incubated at 37 °C for 16–20 h, and the resulting plates were observed for growth and MIC values recorded. The MIC was the lowest concentration of the antibiotic that completely inhibited bacterial growth (visually) within that series of dilutions. For the wells having the negative control of added broth without the emulsion or ciprofloxacin, all wells appeared cloudy due to bacterial growth.

In vitro cell cytotoxicity of the emulsion was evaluated on two human cell lines, human colorectal tumor cells HCT-116, and human embryonic kidney cells HEK 293. HCT-116 were grown in Dulbeco’s Minimum Essential Medium with 10% fetal bovine serum and 0.1% penicillin/streptomycin as complete growth medium for several days at 37 °C under an atmosphere of 5% CO_2_ to reach confluence. HEK 293 cells were grown in Eagle Minimum Essential Medium with 10% fetal bovine serum and 0.1% penicillin/streptomycin as complete growth medium for several days at 37 °C under an atmosphere of 5% CO_2_ to reach confluence. Each cell type was then plated onto 96-well plates, at 50,000 cells per well at a volume of 150 μL with the respective complete growth medium. The cells were counted using a hemocytometer and then incubated for 24 h at 37 °C under an atmosphere of 5% CO_2_. The test emulsion was diluted using the complete growth medium for each cell type and added into the wells of each test plate to give a final concentration of N-acryloylciprofloxacin of 2 mg/mL, 1 mg/mL, 0.5 mg/mL, 0.25 mg/mL, 0.125 mg/mL, and 0.0625 mg/mL within a series. The testing was performed in triplicate, and one well in each triplicate was left untreated as the negative control for 100% growth. The plates were further incubated and monitored for 48 h at 37 °C under an atmosphere of 5% CO_2_. A 5 mg/mL solution of 3-(4,5-dimethyl-2-thiazolyl)-2,5-diphenyltetrazolium bromide (MTT) in sterile phosphate-buffered saline was added to give a 10% final concentration in each well. The plates were then further incubated for 4 h at 37 °C under an atmosphere of 5% CO_2_ to allow for the formation of the purple crystals of 1-(4,5-dimethylthiazol)2-yl)-3,5-diphenylformazan. The liquid was then aspirated from each well and 100 µL of dimethylsulfoxide (DMSO) was added to each well and gently shaken for 1 min to allow for complete dissolution of the crystals. The IC_50_ value for the assay was determined using a BioTek Synergy (BioTek Instruments, Inc., Winooski, VT, USA) H1 hybrid plate reader at both 595 nm and 630 nm. The IC_50_ was determined as the well with at least 50% cell viability compared to the untreated control cell with 100% cell growth.

## 5. Conclusions

Poly (N-acryloylciprofloxacin) nanoparticle emulsions were successfully prepared by modification of the previously reported emulsion polymerization methodology. This is the first report of a polyacrylate nanoparticle emulsion being constructed from solely one monomer derived from an antibacterial agent and far surpasses the current limitation of being able to incorporate only up to 3 weight % (based on the solid content of the polymer) being the acryloylated drug monomer. The average diameter of the nanoparticles in the emulsion is about 930 nm, much larger than the usual poly(butyl acrylate/styrene) constructs, which are generally under 100 nm. The high negative surface charge on these nanoparticles indicates that the emulsions are stable in aqueous media. This method may enable other antibacterial compounds or bioactive agents to be introduced accordingly into the polymeric framework of the emulsified nanoparticle, as a single acryloylated monomer used for the polymerization.

## Figures and Tables

**Figure 1 ijms-26-10034-f001:**
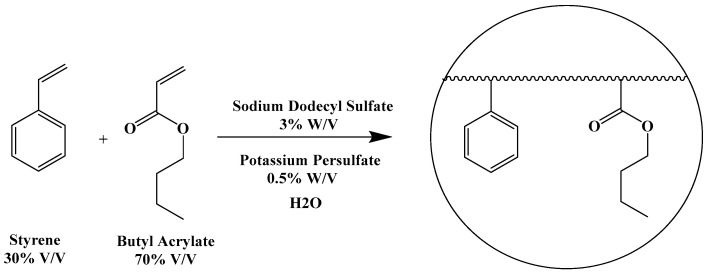
Scheme for emulsion polymerization of butyl acrylate–styrene mixtures.

**Figure 2 ijms-26-10034-f002:**
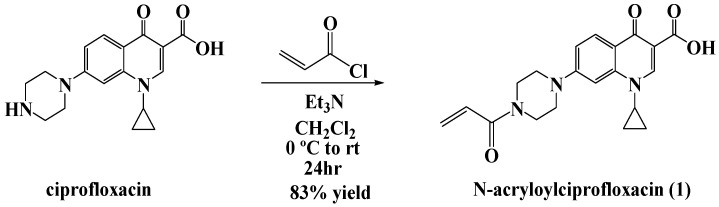
Scheme for the synthesis of N-acryloylciprofloxacin monomer.

**Figure 3 ijms-26-10034-f003:**
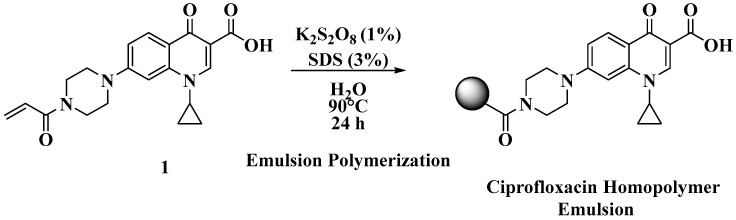
Scheme for preparing poly(N-acryloylciprofloxacin) polymer emulsions.

**Figure 4 ijms-26-10034-f004:**
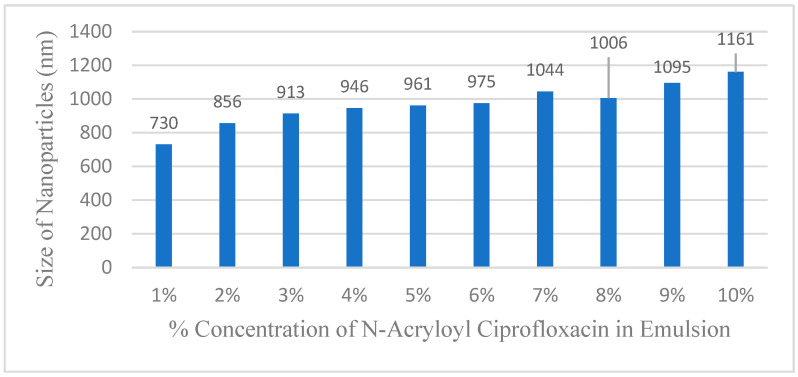
Average size of emulsified nanoparticles versus the % weight concentration of N-acryloylciprofloxacin in the nanoparticle emulsions.

**Figure 5 ijms-26-10034-f005:**
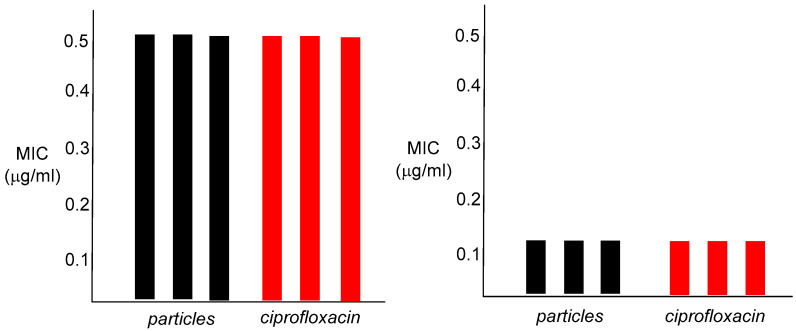
Minimum inhibitory concentration (MIC) values for emulsified poly (N-acryloylciprofloxacin) versus ciprofloxacin hydrochloride. On the left, for *S. aureus*. On the right, for *E. coli.* Each bar represents one assay, with each sample being tested in triplicate.

**Figure 6 ijms-26-10034-f006:**
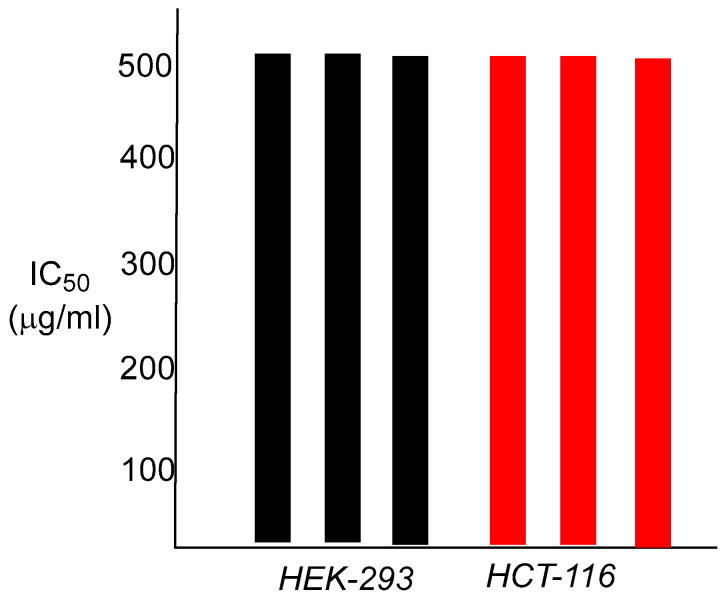
50% cell inhibitory concentration (IC_50_) values for emulsified poly (N-acryloylciprofloxacin) against HEK-293 cells (**left**) and HCT-116 cells (**right**). Values reflect fluorescence intensity at both 595 nm and 630 nm compared to that of the untreated control cell with 100% cell growth. Each bar represents one assay, with each sample being tested in triplicate.

## Data Availability

Data is contained within the article.

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
