# Peer review of "Emulsified Homo (Ciprofloxacin) Polymer Nanoparticles for Antibacterial Applications"

_ijms, 2025, doi:10.3390/ijms262010034_

Round 1
Reviewer 1 Report
Comments and Suggestions for Authors
1. The results of the MIC and MTT assays should be properly plotted and included as complete figures in the manuscript. It is suggested that the appropriate control groups and statistical analyses (such as t-tests) should be included to demonstrate the significance of the antibacterial and cytotoxicity findings.
2. Nanoparticles are typically defined within the 1–100 nm range. However, based on the DLS results, the emulsified nanoparticles appear to be nearly 1 micrometer in size. Since DLS measurements can be affected by aggregation, it is recommended that an additional size characterization technique, such as TEM or SEM, be employed to accurately determine the particle size.
3. The reported particle size of ~970 nm is unusually large for typical drug delivery systems and significantly exceeds the optimal size (<200 nm) for efficient cellular uptake and tissue penetration. It raises concern about whether the drug is released efficiently enough to act intracellularly.
4. While a zeta potential of −63 mV suggests good colloidal stability, the manuscript does not assess long-term emulsion stability or behavior in biological fluids. It is recommended that the authors include a stability study or at minimum discuss whether the emulsions remained stable throughout the biological experiments.
Author Response
The authors are very appreciative to the reviewer for the valuable comments, and we provide these responses.
Comment 1: The results of the MIC and MTT assays should be properly plotted and included as complete figures in the manuscript. It is suggested that the appropriate control groups and statistical analyses (such as t-tests) should be included to demonstrate the significance of the antibacterial and cytotoxicity findings.
Authors' reply: The data we report are indeed preliminary and very limited in scope to a few microbes and cell lines. We agree that a more full assessment to delineate the statistical profiles would be valuable and would enable a better commentary on the biological effects versus suitable controls. We added a note in the manuscript that "we note that these data are preliminary and much more detailed analyses are required for evaluating potential susceptibilities and toxic effects, if any, in appropriate rodent models and additional human cell lines."
Comment 2: Nanoparticles are typically defined within the 1–100 nm range. However, based on the DLS results, the emulsified nanoparticles appear to be nearly 1 micrometer in size. Since DLS measurements can be affected by aggregation, it is recommended that an additional size characterization technique, such as TEM or SEM, be employed to accurately determine the particle size.
Authors' reply: We thank the reviewer for this opinion, which we also share. We were unable to complete the SEM or TEM studies however, due to samples not providing acceptable results for analysis We added this additional commentary to the revised manuscript: "The basis for this 20-fold increase in size is hypothesized to reflect the increasing hydrophilicity of the poly(N-acryloylciprofloxacin) nanoparticle matrix, compared to the highly lipophilic butyl acrylate/styrene polymer chain of the previous emulsified nanoparticles that disfavors the influx of water from the surrounding aqueous environment. Thus, increasing the concentration of the monomer into the polymerization protocol increases the degree in which the resulting nanoparticle matrix draws in water. This is an identical phenomenon with what we previously observed for polyacrylate nanoparticles constructed of hydrophilic glycosylated acrylate monomers, versus those having highly lipophilic hydroxyl-protected glycosylated acrylate monomers. These DLS data afford the average sizes and size distributions of the emulsified particles in aqueous media. We also attempted to evaluate more precisely particle sizes by tunneling electron microscopy and scanning electron microscopy, however, these efforts were thwarted by the deposition of undefined masses rather than individual particles on the wafer surface, even under very high dilution of the emulsion."
Comment 3: The reported particle size of ~970 nm is unusually large for typical drug delivery systems and significantly exceeds the optimal size (<200 nm) for efficient cellular uptake and tissue penetration. It raises concern about whether the drug is released efficiently enough to act intracellularly.
Authors' reply: We added this note to the revised manuscript: "Furthermore, we recognize that the large particle sizes in the emulsion may inhibit the ability to penetrate into cells and tissues and could make the particles susceptible to removal by the lymphatic system before they can effectively control infection. This will have to be examined in much more detail, in due course, if this prototype is deemed useful for further utility in treatment options in animal models. This would have to be evaluated in a future study."
4. While a zeta potential of −63 mV suggests good colloidal stability, the manuscript does not assess long-term emulsion stability or behavior in biological fluids. It is recommended that the authors include a stability study or at minimum discuss whether the emulsions remained stable throughout the biological experiments.
Authors' reply: We agree, and have added this note to the revised manuscript: "Indeed, emulsified samples stored at room temperature have remained completely viable and without significant changes in the DLS data after many months in glass vials."
Reviewer 2 Report
Comments and Suggestions for Authors
This article reports a new method for preparing polyacrylate nanoparticles using N-acryloylciprofloxacin as the sole monomer, which is meaningful for antibacterial applications. The experiments are generally well-designed, but some parts need to be clarified and improved. There are also minor language issues that affect readability.
1 "The procedure entails a free radical-induced emulsion polymerization" should be revised to "The process involves free radical-induced emulsion polymerization".
2 "the emulsions revealed a single population of nanoparticles having" should be revised to "the emulsions showed a single population of nanoparticles with".
3 It is suggested to add more details about why lyophilized powder cannot be re-emulsified , such as possible changes in particle structure.
4 The mechanism of amide bond hydrolysis needs further discussion,and related references need to be cited for support.
5 The size difference between new nanoparticles (970 nm) and previous ones (45-50 nm) should be analyzed with more possible reasons, not just stating "not apparent"
6 Recent articles on the antibio polymer should be cited to update the reference, e.g. 10.1016/j.cclet.2024.109635;
Author Response
We are very appreciative to the reviewer for the valuable comments. We provide the following responses:
Comment 1: "The procedure entails a free radical-induced emulsion polymerization" should be revised to "The process involves free radical-induced emulsion polymerization".
Authors' reply: Wording corrected as suggested.
Comment 2: "the emulsions revealed a single population of nanoparticles having" should be revised to "the emulsions showed a single population of nanoparticles with".
Authors' reply: Wording corrected as suggested.
Comment 3: It is suggested to add more details about why lyophilized powder cannot be re-emulsified , such as possible changes in particle structure.
Authors' reply: Unfortunately we do not have data on this to be able to draw any conclusions. Attempts to reconstitute the dry powders or to prepare highly diluted samples to study individual particle dimensions with TEM or SEM were unsuccessful.
Comment 4: The mechanism of amide bond hydrolysis needs further discussion and related references need to be cited for support.
Authors' reply: Unfortunately we do not have data on the kinetics of amide hydrolysis of the emulsified nanoparticles under the biotesting conditions, to be able to draw any conclusions. The assumption is that the microbes may release the active antibiotic by enzymatic hydrolysis of the N-acryloylciprofloxicin amide linkage, and thus provide for the observed microbiological effects.
Comment 5: The size difference between new nanoparticles (970 nm) and previous ones (45-50 nm) should be analyzed with more possible reasons, not just stating "not apparent"
Authors' reply: We acknowledge this concern and have tried to address this by adding the following analysis: "The basis for this 20-fold increase in size is hypothesized to reflect the increasing hydrophilicity of the poly(N-acryloylciprofloxacin) nanoparticle matrix, compared to the highly lipophilic butyl acrylate/styrene polymer chain of the previous emulsified nanoparticles that disfavors the influx of water from the surrounding aqueous environment. Thus, increasing the concentration of the monomer into the polymerization protocol increases the degree in which the resulting nanoparticle matrix draws in water. This is an identical phenomenon with what we previously observed for polyacrylate nanoparticles constructed of hydrophilic glycosylated acrylate monomers, versus those having highly lipophilic hydroxyl-protected glycosylated acrylate monomers. These DLS data afford the average sizes and size distributions of the emulsified particles in aqueous media. We also attempted to evaluate more precisely particle sizes by tunneling electron microscopy and scanning electron microscopy, however, these efforts were thwarted by the deposition of undefined masses rather than individual particles on the wafer surface, even under very high dilution of the emulsion."
Comment 6: Recent articles on the antibio polymer should be cited to update the reference, e.g. 10.1016/j.cclet.2024.109635;
Authors' reply: The references have been amended with a number of new citations added.
We hope these changes are appropriate and sufficient.
Round 2
Reviewer 1 Report
Comments and Suggestions for Authors
I appreciate the authors’ efforts to revise the manuscript and respond to the reviewers’ concerns. However, several critical issues remain unresolved, and in my view, the work is still too preliminary to warrant publication in IJMS at this stage.
Comment to the authors’ reply on the 1st comment: While the authors acknowledge that the MIC and MTT assay data are preliminary, they did not provide the requested figures with appropriate controls or statistical analyses. As per IJMS policy, transparency and reproducibility of experimental results are essential, and this cannot be addressed by a simple disclaimer.
Comment to the authors’ reply on the 2nd comment: The reported hydrodynamic size of ~970 nm is inconsistent with the definition of nanoparticles and raises important concerns regarding aggregation and the true particle dimensions. Although the authors recognize this issue, no additional characterization (e.g., TEM or SEM) or alternative approach was provided. As a result, the particle size data remain uncertain and the conclusions based solely on DLS measurements are speculative.
Comment to the authors’ reply on the 3rd comment: The unusually large particle size also raises significant doubts about the translational potential of these materials. For drug delivery applications, efficient cellular uptake and tissue penetration typically require particles below 200 nm. Deferring this central issue to “future studies” leaves the manuscript incomplete and weakens the support for the proposed therapeutic relevance.
Taken together, the lack of statistical validation, incomplete particle characterization, and unresolved questions about drug delivery suitability make the current study insufficient for publication. While these findings may provide useful preliminary insights for future work, the manuscript does not yet meet the standards of completeness, rigor, and reproducibility expected for original research articles in IJMS. Therefore, I recommend rejection for the publication in IJMS.
Author Response
Authors' comments: We have carefully read through and amended sections of the manuscript in an attempt to improve the clarity of the wording, especially relating to the description of the research goals. These portions are highlighted in red.
Comment to the authors’ reply on the 1st comment: While the authors acknowledge that the MIC and MTT assay data are preliminary, they did not provide the requested figures with appropriate controls or statistical analyses. As per IJMS policy, transparency and reproducibility of experimental results are essential, and this cannot be addressed by a simple disclaimer.
Authors' response: We added figures for the MIC and MTT data as requested.
Comment to the authors’ reply on the 2nd comment: The reported hydrodynamic size of ~970 nm is inconsistent with the definition of nanoparticles and raises important concerns regarding aggregation and the true particle dimensions. Although the authors recognize this issue, no additional characterization (e.g., TEM or SEM) or alternative approach was provided. As a result, the particle size data remain uncertain and the conclusions based solely on DLS measurements are speculative.
Authors comments: Preparation of samples for the SEM/TEM assays were attempted, but gave no meaningful data. We believe this to be related to the sensitivity of the particles towards hydration and surfactant stabilization in the emulsion, during desiccation under high vacuum in the prep of the wafers. We have done these studies before with much smaller (45 nm) poly(butyl acrylate-styrene) particles, which gave individual particles for SEM and TEM. This does not negate, in our view, the importance of the DLS data we obtained for the aqueous emulsions, indicating that the particles are much larger, and can vary depending on the concentration of the monomer used in the emulsification and radical polymerization.
Comment to the authors’ reply on the 3rd comment: The unusually large particle size also raises significant doubts about the translational potential of these materials. For drug delivery applications, efficient cellular uptake and tissue penetration typically require particles below 200 nm. Deferring this central issue to “future studies” leaves the manuscript incomplete and weakens the support for the proposed therapeutic relevance..... Taken together, the lack of statistical validation, incomplete particle characterization, and unresolved questions about drug delivery suitability make the current study insufficient for publication.
Authors' comments: We openly acknowledge in our discussion of our data and again in the conclusions that these new emulsified polymeric prototypes have properties that may not make them immediately useful for treating infections in humans, or as drug delivery platforms. They may have utility in other applications however, in the antibacterial materials domain, that would need to be determined and proven. That is not part of this study. The significance of this study however was to demonstrate for the first time that polymeric particles made of a single antibacterial monomer within their composition can be formed in aqueous media by emulsion polymerization, and are stable emulsions with antibacterial capabilities. We believe this is significant, in the antibacterial polymer emulsion field, and should be reported.
Reviewer 2 Report
Comments and Suggestions for Authors
accept
Author Response
Thank you.
Round 3
Reviewer 1 Report
Comments and Suggestions for Authors
Thank you to the authors for revising the manuscript and including the requested data. However, the data presentation could be further improved for readability and clarity.
1. For the MIC and MTT data, it is recommended that the results be normalized and that appropriate control experiments be included to better demonstrate the performance of MIC and MTT on the emulsified poly(N-acryloylciprofloxacin). In addition, the number of replicates should be clearly indicated, and error bars should be included in the plots.
2. To further strengthen the analysis, it is suggested that the authors perform statistical testing, such as a one-way ANOVA (analysis of variance), to determine whether the differences among experimental groups are statistically significant. Including p-values in the figures or captions would greatly enhance the robustness of the conclusions.
Author Response
We want to thank this reviewer for the continued interest and input. Here are our responses:
1. For the MIC and MTT data, it is recommended that the results be normalized and that appropriate control experiments be included to better demonstrate the performance of MIC and MTT on the emulsified poly(N-acryloylciprofloxacin). In addition, the number of replicates should be clearly indicated, and error bars should be included in the plots.
Authors' comments: The control experiments we use include a a blank sample (broth only) in each assay for the negative controls and ciprofloxacin HCl as the positive control for the assays. We added a statement for each figure that the assays were done in triplicate for each sample and that the result of each assay is illustrated as a single bar, thus, there is only one measurement per bar, for a total of three bars for each sample tested. All are equivalent, with no variance.
2. To further strengthen the analysis, it is suggested that the authors perform statistical testing, such as a one-way ANOVA (analysis of variance), to determine whether the differences among experimental groups are statistically significant. Including p-values in the figures or captions would greatly enhance the robustness of the conclusions.
Authors' comments: We understand that statistical analysis of data would be needed for sufficiently large sample sizes, but our assays were all done in triplicate per sample with no variance in the values for the MIC or IC50 is identical. Thus, the three assays for the emulsions were uniformly 0.5 ug/ml against S. aureus, as was the positive control ciprofloxacin HCl (0.5 ug/ml=0.5 ug/ml). The same trend was found for all the other assays. For the purposes of our report, we are not aware of any statistical method that provides more detailed clarity about the data being statistically significant than the numbers being compared are uniform and equivalent. If the values were all different and especially if the data set was very large, statistical analysis would certainly be warranted we believe.
Round 4
Reviewer 1 Report
Comments and Suggestions for Authors
Thank you for revising the manuscript. I recommend the manuscript can be accepted for the publication.